# Maternal and demographic factors influencing oral *Candida albicans* in infants: A stratified analysis using a novel partial linear semiparametric mixed-effects model

Sami Leon[1], Nora Alomeir [2], Jin Xiao[2], Tong Tong Wu [1]*

**1** Department of Biostatistics and Computational Biology, University of Rochester Medical Center, Rochester, New York, United States of America, **2** Eastman Institute for Oral Health, University of Rochester Medical Center, Rochester, New York, United States of America

* tongtong_wu@urmc.rochester.edu

## Abstract

*Candida albicans*, a pathogenic fungus implicated in early childhood caries (ECC), plays a crucial role in oral health. While its colonization usually begins at birth, the extent of maternal involvement in yeast transmission to their offspring, particularly across different racial groups, remains unclear. Studies have shown elevated levels of *C. albicans* in both mothers and children, with genetically related fungal strains, suggesting maternal transmission, but the racial component, notably higher levels in Black children, lacks thorough investigation of underlying factors. Our research aimed to address this gap by investigating how maternal and demographic factors such as socioeconomic status and oral health affect *C. albicans* levels in infants across Black and non-Black populations. Employing a partial linear semiparametric mixed-effects model (PLSMM) with variable selection and race-based stratification, we identified predictors that have different effects depending on the infant's race among a large pool of predictors. Through this stratified analysis, we aimed to discern crucial factors significantly contributing to *C. albicans* colonization while minimizing the impact of irrelevant or redundant variables. In this stratified analysis, exclusive breastfeeding ($\beta = -1.06, p < 0.0001$) and maternal marriage ($\beta = -1.09, p = 0.01$) were significant predictors among non-Black infants, while maternal employment ($\beta = -1.55, p = 0.03$) and post-delivery maternal *C. albicans* ($\beta = 1.66, p = 0.049$) were significant among Black infants. Our findings highlighted race-specific associations between *C. albicans* levels in children and factors such as breastfeeding practices, marital status, maternal oral hygiene, and maternal *C. albicans* levels. Our study underscores the importance of race-specific considerations in understanding *C. albicans* colonization in infants, offering insights for tailored interventions and healthcare strategies, particularly for vulnerable populations.

**Data availability statement:** Data cannot be shared publicly as participants did not give consent for their data to be shared in this manner. Since the consent statement approved by the Institutional Review Board (IRB) of the University of Rochester, and signed by the participants, did not include the provision that data would be made publicly available, we do not have participant consent to share this data. Requests for anonymized data can be made to the Institutional Review Board (IRB) of the University of Rochester (https://www.rochester.edu/ohsp/irb-review/).

**Funding:** NIH/NIDCR R01DE031025 and NSF SCC 2238208.

**Competing interests:** The authors have declared that no competing interests exist.

# 1 Introduction

Emerging research evidence suggests a cariogenic potential of oral *Candida* species [1–3]. Intriguing findings have demonstrated a greater prevalence of *Candida albicans* in children with early childhood caries (ECC) than in caries free children [3]. Moreover, a positive association was observed between the carriage of *C. albicans* and cariogenic *Streptococcus mutans* in the saliva and plaque of ECC-affected children [4]. Koo et al. further revealed the synergistic relationship between *C. albicans* and *S. mutans* is facilitated by glucosyltransferases (GTFs) secreted by *S. mutans* [5]. When sucrose is available, the enzyme GtfB latches onto the surface of *C. albicans* cells, leading to the creation of exopolysaccharides (EPS), predominantly insoluble $\alpha$-glucan [6]. EPS plays a pivotal role in fostering the interaction between *C. albicans* and *S. mutans*, thereby encouraging the development of mixed-species biofilms leading to more severe dental caries [5]. The synergistic associations between *C. albicans* and *S. mutans* are associated with exacerbated dental caries in animal models [7] and has a potential indication for a higher rate of caries among children [8].

*C. albicans* oral colonization starts as early as one week [9]. Candida oral colonization is reported to be 45–65% in healthy children and 45% in neonates [10]. A prospective study that aimed to determine the extent of maternal involvement in transmitting *C. albicans* to children, focused on the genetic relatedness of *C. albicans* strains between mother-child dyads and the factors influencing early life colonization [11]. A significant finding was that 94% of mother-child pairs with oral *C. albicans* shared highly genetically related strains, indicating a strong maternal influence on the child's acquisition of this pathogen [11]. The vertical transmission rate of *C. albicans* was reported at 60% (3 out of 5 mother–child pairs) where both the mother and child were carriers of *C. albicans*, indicating a significant role of the mother in transmitting this yeast species to the offspring [12]. Also, one month after birth, 60% of the infants were found to have acquired *C. albicans* through vertical transmission, while the other 40% were affected by horizontal transmission [11].

Oral health conditions such as dental caries and periodontitis in adults exhibit a higher prevalence among Black populations in the United States as compared to White populations [13,14]. Based on the Oral Health Surveillance Report data from 2011 to 2016, the prevalence of ECC is higher in non-Hispanic Black children (28%) compared to non-Hispanic White children (18%) for the age group of 2 to 5 years [14]. This indicates that non-Hispanic Black children have a higher risk of experiencing dental caries in their early years than their non-Hispanic White peers [14]. This information underscores the existence of racial disparities in oral health, particularly in the prevalence of dental caries among young children in the United States. However, the underlying factors contributing to the racial disparity in the context of oral *C. albicans* colonization and carriage in infants and their association with ECC remains unexplored.

In this study, we propose the use of a partial linear semiparametric mixed-effects model (PLSMM) [15] to analyze the concentrations of *Candida albicans* in infants.

The PLSMM enhances both flexibility and interpretability while minimizing specification bias. Unlike linear mixed-effects models (LMM), which impose strict linearity assumptions, this approach, drawing from semiparametric mixed-effects models (SMM), combines the flexibility of nonparametric models with the simplicity of linear regression. By integrating a high-dimensional linear component, the PLSMM addresses dimensionality issues common in nonparametric models. Instead of traditional methods such as splines or kernels, the PLSMM employs a finite dictionary of basis functions, following the approach introduced by Leon and Wu (2025) [15]. This method automatically selects from a diverse range of basis functions, providing a sparse approximation of the nonparametric function. Unlike previous approaches, which may compromise on flexibility and precision, the PLSMM simultaneously tackles issues of dimensionality and flexibility. A detailed implementation of the estimation procedure is provided in the R package `plsmmLasso` [16]. In the work of Leon and Wu (2025) [15], they identified a risk of elevated *C. albicans* levels among Black children compared to non-Black children, as shown in Table 1A. The new stratified analysis here seeks to deepen our understanding by investigating the underlying factors contributing to this racial disparity. Our objective is to explore potential protective or risk factors among Black and non-Black children through a stratified analysis. The stratified analysis aims to help us understand how race-specific factors may interact with *Candida albicans* concentrations in infants, thereby enhancing our understanding of the complex relationship between race, environmental factors, and microbial colonization in early childhood. This comprehensive approach not only sheds light on the determinants of *Candida albicans* colonization but also underscores the importance of considering race as a potential influencing factor in epidemiological studies focused on infant health outcomes.

The dataset to be analyzed consists of 160 mother–infant dyads and includes 86 covariates, tracking children's data at intervals of 1, 2, 4, 6, 12, 18, and 24 months between 11/22/2017 - 08/20/2020, with each child having a minimum of 2 and up to 7 observations. The reported mother-infant dyads were obtained from a parent cohort study that examines the association between oral Candida and early childhood caries onset in children from birth to 2 years of age [17]. Pregnant women and their infants were recruited from patients visiting the University of Rochester Highland Family Medicine or Eastman Institute for Oral Health Perinatal Dental Clinic.

In short, the PLSMM framework enables variable selection, estimation of effect sizes with adjusted *p*-values, modeling of the nonlinear temporal relationship between *Candida albicans* concentration and race, and testing for differences in these functions over time. In this paper, we present a new formulation of PLSMM for stratified analyses.

## 2 Methods

The birth cohort study was approved by the University of Rochester Research Subject Review Board (Protocol #1248). Written informed consent was obtained from all participants, including authorization to access their medical records. For child participants, legal guardians reviewed and signed a consent form for study participation and medical record access. Recruitment occurred between November 2017 and August 2020 at two clinical sites: the University of Rochester Highland Family Medicine (HFM) and the Eastman Institute for Oral Health (EIOH) Perinatal Dental Clinic. The study was concluded in November 2022. To collect relevant data, participants completed self-reported questionnaires covering demographic, socioeconomic, oral health behavior, medical history, and medication use for both mothers and children. This information was verified against the participants' electronic medical records. The research team had access to identifiable participant data throughout and following the data collection phase.

### 2.1 Partial Linear Semiparametric Mixed-Effects Model (PLSMM)

We apply partial linear semiparametric linear mixed-effects model (PLSMM) proposed by Leon and Wu (2025) [15] to analyze the impact of race on *Candida albicans* (Ca) concentration, employing nonlinear time effects for each race group using distinct basis functions, and conducting variable selection to identify potential variables linked to Ca concentration. The data were collected during 11/22/2017 - 08/20/2020.

The PLSMM is defined as follows:

$$Ca(t) = X\beta + f_g(t) + Z\phi + \epsilon,$$

where $t$ is the time (in days), $g$ is the grouping variable (0 for non-Black and 1 for Black), $\phi$ is the random intercept for individual subjects, $Z = \text{diag}(Z_1, ..., Z_N)$ with $Z_i$ a vector of $n_i$ ones, and $\epsilon$ is the random noise. The regression coefficients $\beta$ with its associated design matrix $X$ capture the effects of the covariates on $Ca$, and $f_g(t)$ represents the semiparametric part of the model which captures the effect of time for each infant in each group. The function $f_g$ is unknown and needs to be estimated. The random terms $\phi_i$ and $\epsilon_{ij}$ are assumed to have zero conditional means such that $E[\phi_i|X_i, t_i] = 0$ and $E[\epsilon_{ij}|X_i, t_i] = 0$ with finite variances where $\text{Var}[\phi_i] = \sigma_\phi^2$ and $\text{Var}[\epsilon_{ij}] = \sigma^2$.

**Remark 1** (Identifiability). *To retain identifiability, we use a mean-zero constraint enforcing both nonlinear functions to have a mean of zero, expressed as $E[f_0] = E[f_1] = 0$. Details about the constraint can be found in [15].*

## 2.2 Estimation of PLSMM

The nonlinear function $f_g$ is modeled using a sparse linear combinations of elements in a "dictionary", which contains a wide range of functions $\{\psi_1, ..., \psi_M\}$. The approximation is then given by

$$\tilde{f}_{\alpha,g} = \begin{cases} \sum_{k=1}^{M} \alpha_k \psi_k & \text{for } g = 0, \\ \sum_{k=1}^{M} \alpha_{M+k} \psi_k & \text{for } g = 1, \end{cases}$$

where $\alpha = (\alpha_1, ..., \alpha_{2M})^\top$ is a sparse vector of coefficients estimated using lasso. The dictionary can be a collection of basis functions from different bases (e.g., splines with fixed knots, Fourier basis, power functions, etc.). For the ease of notation and without loss of generality, we assume the observations are sorted by groups. The approximation $\tilde{f}_{\alpha,g}$ can be written using the following block matrix

$$\Psi = \begin{pmatrix} \Psi_0 & 0 \\ 0 & \Psi_1 \end{pmatrix},$$

such that $\tilde{f}_{\alpha,g} = \Psi\alpha$, where $\Psi_0 = (\psi_k(t_{ij}))_{k,i,j}$ with $t_{ij}$ being the time points at the $j$th observation of subject $i$ in group 0, similarly $\Psi_1 = (\psi_k(t_{ij}))_{k,i,j}$ with $t_{ij}$ being the time points in group 1. We'd like to point out that the matrices $\Psi_0$ and $\Psi_1$ do not need to be estimated using the same dictionary, i.e., different sets of basis functions can be used for each search. Thus, $\Psi$ allows to have two distinct nonlinear functions with possibly a very different shape for each group. The first $M$ coefficients of $\alpha$ are associated with group 0, the coefficients from $M+1$ to $2M$ are associated with group 1. The model can then be written as follows:

$$Ca(t) = X\beta + \Psi\alpha + Z\phi + \epsilon. \tag{1}$$

The nonparametric component $\Psi\alpha$ allows for a distinct nonlinear relationship between the longitudinal response and time for each group. This approach allows the shape of the functions to be determined by the data rather than assumed a priori. A rich dictionary provides flexibility, while sparsity in $\alpha$ selects only the most relevant basis functions from the dictionary.

To estimate the parametric and nonparametric parts of model (1), collectively denoted by $\theta = (\beta, \alpha, \sigma_\phi^2, \sigma^2)$, we adopt a penalized maximum likelihood framework.

The model parameters are estimated using a penalized EM algorithm that maximizes a penalized version of the conditional expectation of the complete-data log-likelihood of $\boldsymbol{Ca}(\boldsymbol{t})$. Specifically, the algorithm maximizes 2 with respect to the parameter vector $\theta$.

$$\log L_1(\boldsymbol{Ca}(\boldsymbol{t})|\boldsymbol{\phi};\theta,\sigma^2) + \log L_2(\boldsymbol{\phi};\sigma_\phi^2) - 2\sigma^2\sqrt{\frac{\gamma \log M}{n}} \sum_{k=1}^{2M} ||\psi_k|| |\alpha_k| - \lambda \sum_{k=1}^{p} \beta_k, \tag{2}$$

with $||\psi_k||$ is the Euclidean norm such that $||\psi_k||^2 = \frac{1}{n} \sum_{i=1}^{N} \sum_{j=1}^{n_i} \psi_k^2(t_{ij})$, $\log L_1(\boldsymbol{Ca}(\boldsymbol{t})|\boldsymbol{\phi};\theta,\sigma^2) = -\frac{1}{2n}\log(2\pi\sigma^2) - \frac{1}{2\sigma^2}||\boldsymbol{Ca}(\boldsymbol{t}) - \boldsymbol{X\beta} - \boldsymbol{\Psi\alpha} - \boldsymbol{Z\phi}||^2 - \frac{1}{2\sigma^2}\text{Tr}(\boldsymbol{Z}\text{Var}_\theta(\boldsymbol{\phi}|\boldsymbol{Ca}(\boldsymbol{t}))\boldsymbol{Z}^\top)$ and $\log L_2(\boldsymbol{\phi};\sigma_\phi^2) = -\frac{1}{2N}\log(2\pi\sigma_\phi^2) - \frac{1}{2\sigma_\phi^2}\boldsymbol{\phi}^\top\boldsymbol{\phi} - \frac{1}{2\sigma_\phi^2}\text{Tr}(\text{Var}_\theta(\boldsymbol{\phi}))$. The detailed mathematical derivation of the estimation procedure is provided in [15], where the algorithm was developed for a general variance–covariance matrix $\Sigma_\phi$. Here, we focus on the special case assuming $\text{Var}[\phi_i] = \sigma_\phi^2$, which leads to simpler expressions for the likelihood terms. Using the EM algorithm to maximize 2 we obtain the estimators of the model parameters, denoted $\hat{\theta} = (\hat{\boldsymbol{\beta}}, \hat{\boldsymbol{\alpha}}, \hat{\sigma}_\phi^2, \hat{\sigma}^2)$. The estimation procedure can be carried out using the package `plsmmLasso` [16], where the function `plsmm_lasso()` is used for parameter estimation.

The model estimation depends on the value of the tuning parameters $\gamma$ and $\lambda$. We select these parameters by performing a grid search and then selecting the model that minimize the Bayesian Information Criterion (BIC) [18]. The BIC is a method that allows to compare the relative goodness of fit of different models. It is based on Bayesian statistics and attempts to balance the model fitting to the data and the model complexity by penalizing models with a large number of parameters such that

$$\text{BIC} = -2l(\theta) + k\log(n)$$

where $l(\theta)$ is the likelihood of the model with parameters $\theta$, $k$ the number of non-zero elements in $\boldsymbol{\beta}$ and $n$ is the number of observations. A pseudocode outlining the complete estimation procedure is presented in [15] and the tuning can be performed using the `tune_plsmm()` function from the R package `plsmmLasso` [16].

The model can be interpreted similarly to a mixed-effects model, with some additional features. The coefficients $\boldsymbol{\beta}$ represent the effects of the covariates, $\sigma_\phi^2$ captures the correlation between repeated measurements, and $\alpha$ allows the model to approximate the true underlying nonlinear functions of time. The model's goals are to estimate and test differences in the nonlinear time trends between groups and accurately estimate the covariate effects through $\boldsymbol{\beta}$ while performing variable selection among these covariates. In the next two sections, we outline the procedures used to carry out inference under this model.

## 2.3 Post-selection inference

Post-selection inference is used to provide valid statistical inference that accounts for the uncertainty introduced in the variable selection step [19]. We use the debiasing procedure proposed in the paper by Leon and Wu (2025) [15].

First, the data requires transformation. Let $\tilde{\boldsymbol{y}}_t = \boldsymbol{Ca}(\boldsymbol{t}) - \boldsymbol{\Psi}\hat{\boldsymbol{\alpha}}$, the remainder of the responses with the estimated nonlinear component removed. The estimate of the covariance of the random components is: $\hat{\Sigma} = \hat{\sigma}_\phi \boldsymbol{Z}\boldsymbol{Z}^\top + \hat{\sigma}^2 \boldsymbol{I}$. Let $\boldsymbol{X}_t$ and $\boldsymbol{y}_t$ denote the transformed observations such that $(\boldsymbol{X}_t, \boldsymbol{y}_t) = (\hat{\Sigma}^{-1/2}\boldsymbol{X}, \hat{\Sigma}^{-1/2}\tilde{\boldsymbol{y}})$. The debiased estimate of $\hat{\beta}$ is defined by

$$\hat{\beta}_j^d = \hat{\beta}_j + \frac{\hat{\boldsymbol{w}}_j^\top(\boldsymbol{y}_t - \boldsymbol{X}_t\hat{\boldsymbol{\beta}})}{\hat{\boldsymbol{w}}_j^\top(\boldsymbol{X}_t)_{.j}},$$

where $\hat{\boldsymbol{w}}_j$ is a correction score, which can be calculated using either an additional lasso regression or a quadratic optimization method. Here we take the lasso approach using the transformed data. Define the correction score

$\hat{w}_j = (X_t)._{.j} - (X_t)._{.,-j}\hat{\kappa}_j$, where $(X_t)._{.j}$ is the $j$-th column of the transformed matrix $X_t$ and $(X_t)._{.,-j}$ is all columns of $X_t$ except for the $j$-th one and

$$\hat{\kappa}_j = \underset{\kappa_j}{\operatorname{argmin}} \left( ||(X_t)._{.j} - (X_t)._{.,-j}\kappa_j||_2^2 + \lambda_j||\kappa_j||_1 \right) \tag{3}$$

with the tuning parameter $\lambda_j > 0$.

The two-sided confidence interval at a $100 \times (1-\alpha)\%$ confidence level for $\beta_j$ can be constructed as:

$$\hat{\beta}_j^d \pm z_{\alpha/2}\sqrt{\hat{V}_j},$$

where $z_{\alpha/2}$ is the $(\alpha/2)$th quantile of a standard normal distribution and $\hat{V}_j$ is an estimator of the variance of $\hat{\beta}_j^d$ that can be estimated using the following empirical variance estimate

$$\hat{V}_j = \frac{\sum_{i=1}^{n} \left[ (\hat{w}_j^i)^\top (y_t^i - X_t^i\hat{\beta}) \right]^2}{(\hat{w}_j^\top (X_t)._{.j})^2}$$

with $\hat{w}_j^i$ being the $i$th sub-vector of $\hat{w}_j = ((\hat{w}_j^1)^\top, ..., (\hat{w}_j^n)^\top)^\top$, and $y_t^i$ is the $i$th sub-vector of $y_t$. Post-selection inference is necessary to correct the bias introduced by the variable selection procedure. The debiasing procedure corrects for this bias by first transforming the data to remove correlations induced by the random effects, resulting in a version of the dataset $(X_t, y_t)$ that is suitable for debiasing. Using these transformed observations, the original coefficient estimates are then adjusted to account for the bias introduced by the selection procedure. This approach ensures that the resulting estimates more accurately reflect the true effects of the predictors and allows for the construction of valid confidence intervals for the fixed-effect coefficients. The debiasing procedure can be carried out using the `debias_plsmm()` function from the R package `plsmmLasso` [16].

## 2.4 Testing of the nonparametric component

We first focus on performing an overall test to compare the nonlinear functions between two groups. This hypothesis is:

$$H_0 : f_0 = f_1 \quad \text{vs.} \quad H_1 : f_0 \neq f_1.$$

To quantify the difference, we consider the $L_2$ norm of the difference between the two nonlinear functions $f_0$ and $f_1$. This norm tends to be small when the null hypothesis $H_0$ holds true, and conversely, when $H_0$ is not true the $L_2$ norm will be relatively large. We propose using the following test statistic

$$T = ||f_0(t) - f_1(t)||^2 = \int (f_0(t) - f_1(t))^2 dt.$$

The observed test statistic is then calculated by replacing $f_0$ and $f_1$ by their estimated function $\hat{f}_0(t)$ and $\hat{f}_1(t)$. The $L_2$ norm-based tests have previously been used in similar contexts such as [20]. Intuitively, this test defines a metric that quantifies the overall difference between the two estimated functions: if the functions are similar, the metric is small, whereas larger values indicate greater differences.

The distribution of this $L_2$ norm-based test statistic is generally unknown. In order to assess its statistical significance, we employ a bootstrapping method as described in the paper by Leon and Wu (2025) [15].

Frequently, there is also interest in constructing joint confidence intervals to assess the difference between functions at a specific time point, say $f_1(t^*) - f_0(t^*)$. Following the methodology presented in the work by Leon and Wu (2025) [15], we construct bootstrapped joint confidence intervals based on $B$ bootstrap samples, given by

$$\bar{d}(t^*) \pm q_{1-\alpha/2} \bar{s}(t^*),$$

where $q_{1-\alpha/2}$ is the $(1 - \alpha/2)$th quantile of $M_b, b = 1, ..., B$ with

$$M_b = \max_t \frac{|d^b(t) - \bar{d}(t)|}{\bar{s}(t)}.$$

Here $d^b(t) = \hat{f}_1^b(t) - \hat{f}_0^b(t)$, $\bar{d}(t) = \sum_{b=1}^B d^b(t)/B$ and $\bar{s}^2(t) = \sum_{b=1}^B (d^b(t) - \bar{d}(t))^2/B$ and $\hat{f}_1^b, \hat{f}_0^b$ the estimated function for bootstrap sample $b$.

Conceptually, $\bar{d}(t^*)$ represents the average estimated difference between the two functions across the bootstrap samples at time point $t^*$. The quantity $M_b$, defined as the maximum standardized deviation across all time points, ensures that the resulting joint confidence interval captures the uncertainty over the entire time range simultaneously. By taking the maximum, the interval maintains the nominal coverage probability for all points along the continuum, effectively accounting for multiple comparisons across time. Both tests can be conducted using the `test_f()` function from the R package `plsmmLasso` [16].

## 3 Results

### 3.1 Data description

The study sample consists predominantly of low-income patients, with 54% of the children identified as Black. The demographic comparison between Black and non-Black mothers reveals several key patterns, as detailed in Table 1.

Maternal age and rates of college education were similar across the two groups. However, Black mothers had higher employment rates and lower marriage rates than non-Black mothers. A greater proportion of non-Black mothers reported not brushing their teeth on the daily basis. Clinical oral health indicators were comparable between groups, including the proportions with plaque index scores $\geq 2$ and salivary Candida level 3. Post-delivery Candida infections occurred at similar rates, though oral anti-fungal treatment was less common among Black mothers. Furthermore, the proportion of male infants was higher in the Black group (54% vs 45%).

**Table 1**. Descriptive comparison between Black and non-Black demographics.

|  | Black (n = 85) | Non-Black (n = 75) |
|---|---|---|
| Mother Mean Age (years) | 27.0 | 28.1 |
| Infant Male | 46 (54%) | 34 (45%) |
| Mother College Education | 21 (25%) | 18 (25%) |
| Mother Employed | 48 (57%) | 32 (42%) |
| Mother Married | 16 (19%) | 19 (25%) |
| Mother Smoking | 9 (11%) | 17 (23%) |
| Mother Not Brush Teeth Daily | 4 (5%) | 7 (9%) |
| Mother Plaque Index ($\geq$ 2) | 31 (37%) | 31 (41%) |
| Mother Saliva Ca Level3 | 21 (25%) | 22 (29%) |
| Mother Post Delivery *Candida* | 37 (44%) | 32 (42%) |
| Mother Oral Antifungal | 4 (5%) | 7 (9%) |

## 3.2 Stratified analysis

The overall trends of our model can be seen in Fig 1. These trends exhibit a similar pattern across groups: a rapid increase followed by a decrease, succeeded by a gradual rise and then a slow decline. The main distinction lies in the magnitudes of these changes. Black children start at a slightly higher baseline value, but their increases are larger, consistently placing their trend above the non-Black group.

The effect of time can be seen in the second plot, where the trends mostly overlap, with the greatest difference observed at month 1 and 2. To facilitate hypothesis testing of the nonlinear functions, we conducted 10,000 bootstrap resamples, generating corresponding estimated trajectories for each group. From these resampled trajectories, joint confidence bands were constructed to test the difference between $f_{Black}$ and $f_{Non-Black}$, as presented in the final plot of Fig 1. The joint confidence bands for the differences between the nonlinear functions shows no significant difference at the observed time points; all the confidence bands for each time points largely cover 0. Additionally, an overall test

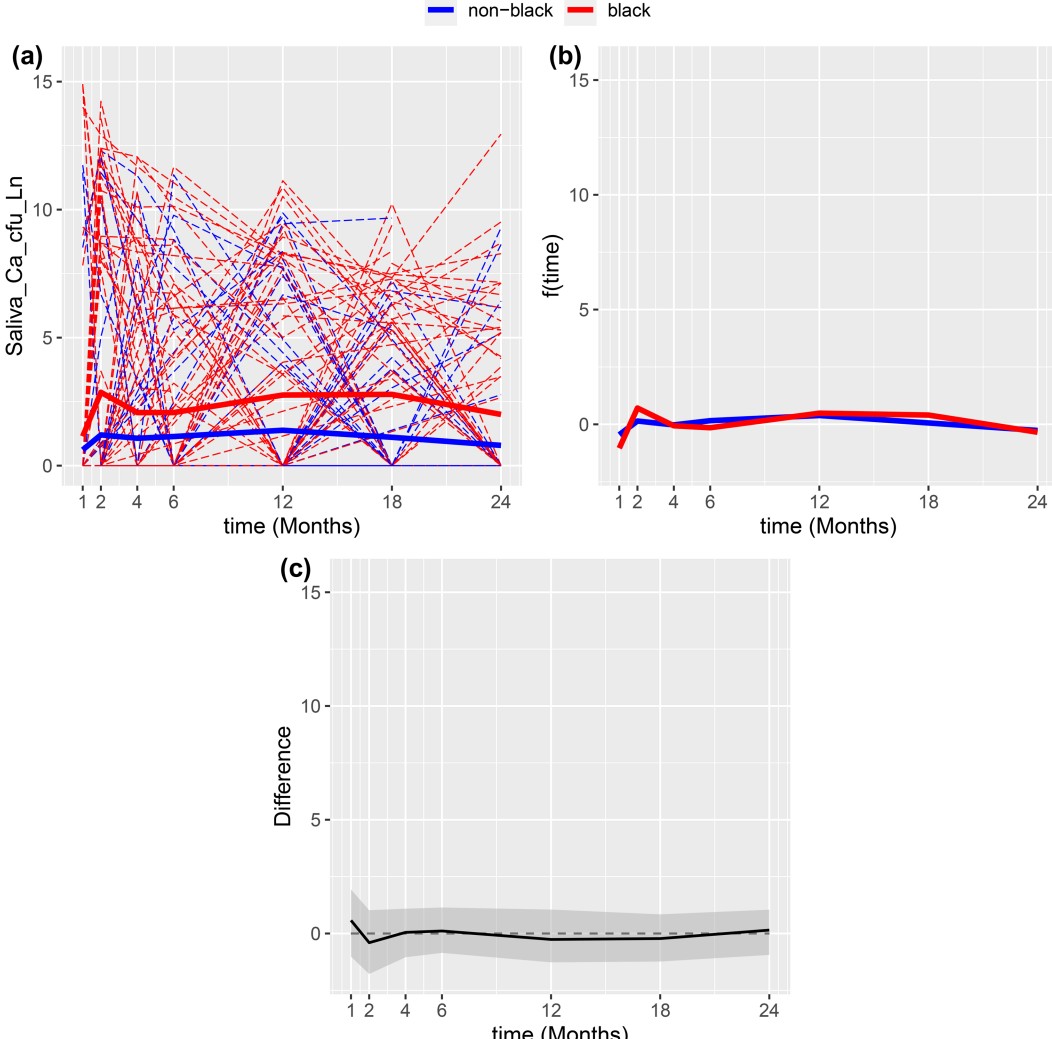

**Fig 1.** (a) Sample (dash curves) and estimated overall trajectories (solid curves) for Black (red) and non-Black (blue) groups; (b) estimated nonlinear functions; (c) the estimate of $f_{Black}(t) - f_{non-Black}(t)$ (Black) and its joint confidence intervals (shaded gray).

comparing these nonlinear functions yields a *p*-value of 0.49, suggesting insufficient evidence to claim that the time trends differ significantly between the groups.

To conduct a stratified analysis, we create a design matrix allowing for the estimation of independent coefficients for each subgroup, or "stratum", defined by the categorical variable race. The results of the stratified analysis, presented in Table 2, reveal important factors associated with the concentration of *C. albicans*. Four predictors reached statistical significance. The interpretation of the remaining six, which did not reach significance, is therefore descriptive in nature. Infants with non-Black mothers who exclusively breastfeed or are married have significantly lower levels of *C. albicans*, indicating a protective effect against *Candida* colonization. Conversely, infants of Black mothers with detected *Candida* status post-delivery show increased *C. albicans* concentration, suggesting a risk factor, however this risk factor is not observed for non-Black children. What makes this finding interesting is that the likelihood of both non-Black and Black mothers having *Candida* detected post-delivery is similar, a test was conducted and the observed difference was not significant. One possible explanation for this discrepancy could be attributed to vertical transfers. It's plausible that such transfers are more prevalent in the case of Black mothers, leading to an increased transmission of *Candida albicans* to their infants.

There's no significant effect between *C. albicans* concentration and infants of Black mothers with a college-level education or higher, although this could be protective. Factors like Black mothers not brushing their teeth daily, receiving oral antifungal prescriptions within six months post-pregnancy, or having a saliva *C. albicans* level of 400 or higher aren't statistically significant but could pose risks. The impact of race alone isn't statistically significant, but Black infants tend to have a larger *C. albicans* concentration of 0.49.

Although some coefficients lack statistical significance individually, they were all selected by the lasso procedure, enhancing the model's predictive accuracy. This implies that while certain variables may not reach traditional significance thresholds on their own, their inclusion in the model improves its ability to predict *C. albicans* concentration accurately. It is not unexpected that race did not reach statistical significance in the stratified analysis, even though it was a strong predictor in the non-stratified model (Table 1A, *p* = 0.0002). The stratified analysis decomposes the racial effect into contributions from other predictors, effectively accounting for potential confounding and mediation by socio-economic, environmental, and genetic factors. As a result, the direct effect of race is attenuated, indicating that race itself is unlikely to be the primary driver of elevated *C. albicans* levels in Black infants. Instead, the observed racial disparities appear largely mediated or confounded by other variables included in the model. The interpretation of the race coefficient in this context encapsulates the residual impact of race, elucidated after accounting for its interaction with other predictors within our model. It denotes the specific influence of race that remains unexplained by the encompassing set of predictors considered.

**Table 2**. **Post-selection inference for the real data application.** The estimates denote the coefficients derived from our lasso procedure, while the debiased coefficients represent the values obtained after employing an adapted debiasing method which allows to compute *p*-values and confidence intervals.

| Variable | Orig. Coef. | Debias Coef. | *p*-value | CI lowe | CI upper |
|---|---|---|---|---|---|
| Non-Black Exclus. Breastfeed | −0.59 | −1.06 | <0.0001 | −1.56 | −0.57 |
| Non-Black Mother Married | −0.34 | −1.09 | 0.01 | −1.92 | −0.26 |
| Black Mother Employed | −0.60 | −1.55 | 0.03 | −2.95 | −0.15 |
| Black Mother Post Delivery *Candida* | 0.68 | 1.66 | 0.049 | 0.01 | 3.31 |
| Black Mother Education (College) | −0.05 | −0.68 | 0.22 | −1.78 | 0.42 |
| Black Mother Plaque Index ($\geq 2$) | 0.18 | 0.75 | 0.22 | −0.46 | 1.96 |
| Black Mother Not Brush Teeth Daily | 0.98 | 2.26 | 0.24 | −1.52 | 6.04 |
| Black Mother Oral Antifungal | 0.75 | 2.33 | 0.31 | −2.19 | 6.85 |
| Mother Saliva Ca level3 | 0.86 | 0.66 | 0.55 | −1.50 | 2.72 |
| Race (Black) | 0.49 | 0.37 | 0.60 | −1.03 | 1.77 |

## 4 Discussion

Relatively little was known about the relationship between race and ethnicity and oral candidiasis. Our study investigated maternal factors that affect the carriage level of infant oral *C. albicans*, a fungus linked to early childhood tooth decay, with a particular attention to racial differences since Black children show higher infection rates. Using advanced statistical modeling, we analyzed how maternal and demographic factors influence *C. albicans* carriage differently across Black and non-Black populations. The key findings revealed race-specific patterns in what protects infants from having a higher amount oral *C. albicans*. For non-Black infants, exclusive breastfeeding ($\beta = -1.06, p < 0.0001$) and maternal marriage ($\beta = -1.09, p = 0.01$) were strongly protective factors. In contrast, for Black infants, maternal employment ($\beta = -1.55, p = 0.03$) served as a protective factor, while having a mother with post-delivery Candida infection increased risk ($\beta = 1.66, p = 0.049$).

Our research found a notably higher prevalence of *C. albicans* in Black infants, aligning with the observations by Jenks et al. that Black individuals are more susceptible to both surface-level and invasive *Candida* infections [21]. The researchers highlighted that the exact causes of this increased susceptibility remain uncertain, suggesting it might arise from broader social health determinants, healthcare access inequalities, and various socioeconomic conditions, rather than strictly genetic or biological reasons [21]. Our study suggests that for Black infants maternal employment might protect infants against the transmission of *C. albicans* from mothers to infants, possibly due to better healthcare access, hygiene practices, and reduced mother-infant contact frequency, minimizing vertical transmission opportunities. Although *C. albicans* detection among women during the postpartum stage is consistent across racial groups, its presence in Black mothers significantly raises the risk of passing it to infants from close mother-infant contact, such as through breastfeeding. We have analyzed the genetic relatedness of *C. albicans* isolated from both mothers and children and indicated the high rate of sharing of genetically identical/highly related strains [11]. Although previous studies emphasized the role of poor maternal oral hygiene in increasing *C. albicans* transmission from mother to child, we did not observe any racial differences in plaque index, which is an indication of the effectiveness of oral hygiene. Additionally, horizontal transmission sources like daycare also pose a risk for *C. albicans* acquisition in infants, showing that both the transmission point and contact frequency are crucial in *C. albicans* transmission dynamics [11]. Our research highlighted a significantly high Candida risk Postpartum, in infants born to Black mothers. Due to disparity driven by social determinants, especially in communities where factors like crowded living conditions, limited access to high-quality postpartum care, and other socioeconomic challenges are more prevalent, infants are more likely to be exposed to infectious agents after delivery [22]. In addition, some mothers may continue to carry Candida after delivery, which can be transferred through vertical transmission by kissing, breastfeeding, and other means. A systematic review by Jang et al. found that pregnant women in their third trimester have higher salivary Candida carriage compared to non-pregnant women [23].

Among non-Black infants breastfeeding showed significant protective response against the carriage of *C. albicans*. Possible explanation lies in: a) breast milk contains a rich supply of antibodies (especially IgA), immune cells, and other factors that play a critical role in shaping the infant's immune system; these components can protect infants from infections and diseases by directly fighting pathogens and modulating the infant's immune responses [24]; b) compared to some formula milk, breast milk has a lower risk of leading to excessive sugar exposure. Excessive sugar intake from early life can lead to a higher risk of obesity, diabetes, and other metabolic diseases, indirectly affecting the child's immune health [25]. Consequently, the elevated risk of caries and colonization by *C. albicans* in Black infants can be attributed to the lower rates of breastfeeding initiation and duration observed among Black mothers, in comparison to mothers from other racial or ethnic groups [26]. Furthermore, married mothers may experience protective benefits due to a combination of factors, including financial stability, social support, and legal protections associated with marriage. This can contribute to better outcomes for children and the family as a whole [27]. In our study, infants with married non-Black mothers had significantly lower levels of *C. albicans*, indicating a protective effect against *Candida* colonization.

 

Black, Hispanic, and other minority women in the United States continue to have lower breastfeeding rates compared to white women. The disparities are influenced by various factors, including socioeconomic status, education level, cultural beliefs, and access to healthcare and breastfeeding support [26]. Black women have the lowest rates of breastfeeding initiation and continuation at 6 and 12 months compared to all other racial/ethnic groups [26]. Consistent with these findings, we observed a lower prevalence of exclusive breastfeeding among Black infants. The lack of data or underrepresentation of purely breastfed Black infants in our study may explain why exclusive breastfeeding was not statistically significant for Black infants. Given that the protective benefits of exclusive breastfeeding are expected to apply across all racial groups, without suggesting a race-specific effect, another possible explanation for the lack of significance could be the potential for vertical transmission being more common among Black mothers. While exclusive breastfeeding provides numerous benefits, the close contact involved in breastfeeding may inadvertently increase the risk of vertical transmission. This is reflected in the debiased coefficients for exclusive breastfeeding, which are -0.59 for Black mothers (not statistically significant) compared to -1.06 for White mothers (statistically significant).

Utilizing the PLSMM framework enables us to achieve several key objectives: (1) perform variable selection to identify potential associations with *Candida albicans* concentration in each stratum; (2) estimate the effect sizes and obtain adjusted *p*-values for the selected variables; (3) model the relationship between the *Candida albicans* concentration and race over time; (4) model the nonlinear pattern of the temporal effects; (5) test for differences in the nonlinear functions globally and at specific time points. In a PLSMM, the longitudinal response can be considered as a function of time, where time is treated as a continuous variable. The model fitting is data-driven and has no restriction on the shape of the fitted model. By doing that, a nonlinear relationship is allowed between the longitudinal response and predictors. It is particularly advantageous for this application as the function of time appears to be nonlinear and non-smooth. Alternative approaches of PLSMM generally assume a certain degree of smoothness in the underlying function, and if the true function is not smooth, the estimators may fail to accurately capture the underlying structure. PLSMM adds flexibility to the model specification and has demonstrated superiority over splines in scenarios where the nonparametric component involves both smooth and non-smooth functions [28].

Overall, *C. albicans* carriage differences between Black and non-Black infants reflect complex transmission pathways within distinct social contexts rather than isolated factors. This requires an integrated framework positioning social determinants as foundational to both vertical transmission (pregnancy/delivery) and horizontal transmission (post-birth exposure). Race-specific protective factors support this model: exclusive breastfeeding and mother's marriage status protect non-Black infants, suggesting traditional structures operate effectively within their context. Conversely, maternal employment protects Black infants while post-delivery maternal Candida increases infants' risk. These patterns indicate different social environments create distinct risk and protective mechanisms through both transmission pathways, requiring targeted interventions addressing population-specific circumstances.

Given these findings, further research into the role of vertical and horizontal transmission is necessary. Such studies could provide deeper insights into how environmental factors, childcare practices, and community health behaviors contribute to the spread of *C. albicans* among infants and children. Understanding these dynamics is crucial for developing comprehensive prevention strategies that encompass both pathways of transmission.

## Appendix A. Non-stratified analysis results

**Table 3. Estimates of regression coefficients using the adaptive debiasing method.** Column 1 shows the estimates using the penalized-EM algorithm, and Column 2 lists the debiased estimates, and Columns 3-5 report the debiased *p*-value and 95% confidence intervals.

| Variable | Orig. Coef. | Debias Coef. | *p*-value | CI lower | CI upper |
|---|---|---|---|---|---|
| Race (Black) | 1.33 | 1.48 | 0.0002 | 0.69 | 2.27 |
| Mother Employed | −0.52 | −1.33 | 0.001 | −2.14 | −0.52 |
| Exclus. Breast feed | −0.43 | −0.97 | 0.009 | −1.70 | −0.24 |
| Mother Post Delivery *Candida* | 0.56 | 1.13 | 0.02 | 0.17 | 2.08 |
| Perinatal Dental Treatment (pre/post) | −0.41 | −0.99 | 0.02 | −1.78 | −0.19 |
| Mother Saliva Ca Level3 | 0.56 | 0.68 | 0.29 | −0.58 | 1.93 |
| Mother Education (College) | −0.14 | −0.37 | 0.43 | −1.28 | 0.55 |

# Author contributions

**Conceptualization:** Sami Leon, Tong Tong Wu.

**Data curation:** Nora Alomeir.

**Formal analysis:** Sami Leon.

**Funding acquisition:** Jin Xiao, Tong Tong Wu.

**Methodology:** Sami Leon, Tong Tong Wu.

**Project administration:** Jin Xiao, Tong Tong Wu.

**Resources:** Jin Xiao, Tong Tong Wu.

**Software:** Sami Leon.

**Supervision:** Tong Tong Wu.

**Writing – original draft:** Sami Leon, Nora Alomeir, Jin Xiao, Tong Tong Wu.

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
