## [Decision Letter · Decision Letter 0]

9 Oct 2025

PONE-D-25-38257

Maternal and Demographic Factors Influencing Oral Candida albicans in Infants: A Stratified Analysis Using a Novel Partial Linear Semiparametric Mixed-effects Model

PLOS ONE

Dear Dr. Wu,

Thank you for submitting your manuscript to PLOS ONE. After careful consideration, we feel that it has merit but does not fully meet PLOS ONE’s publication criteria as it currently stands. Therefore, we invite you to submit a revised version of the manuscript that addresses the points raised during the review process.

We look forward to receiving your revised manuscript.

Kind regards,

Geelsu Hwang, Ph.D.

Academic Editor

PLOS ONE

Journal Requirements:

“NIH/NIDCR R01DE031025 and NSF SCC 2238208”

3. In the online submission form you indicate that your data is not available for proprietary reasons and have provided a contact point for accessing this data. Please note that your current contact point is a co-author on this manuscript. According to our Data Policy, the contact point must not be an author on the manuscript and must be an institutional contact, ideally not an individual. Please revise your data statement to a non-author institutional point of contact, such as a data access or ethics committee, and send this to us via return email. Please also include contact information for the third party organization, and please include the full citation of where the data can be found.

a) If there are ethical or legal restrictions on sharing a de-identified data set, please explain them in detail (e.g., data contain potentially identifying or sensitive patient information, data are owned by a third-party organization, etc.) and who has imposed them (e.g., a Research Ethics Committee or Institutional Review Board, etc.). Please also provide contact information for a data access committee, ethics committee, or other institutional body to which data requests may be sent

5. In the online submission form you indicate that your data is not available for proprietary reasons and have provided a contact point for accessing this data. Please note that your current contact point is a co-author on this manuscript. According to our Data Policy, the contact point must not be an author on the manuscript and must be an institutional contact, ideally not an individual. Please revise your data statement to a non-author institutional point of contact, such as a data access or ethics committee, and send this to us via return email. Please also include contact information for the third party organization, and please include the full citation of where the data can be found.

“Leon, Alomeir, Xiao and Wu were partly supported by NIH/NIDCR R01DE031025 and NSF SCC 2238208.”

“NIH/NIDCR R01DE031025 and NSF SCC 2238208”

7. Please upload a copy of Figure 1, to which you refer in your text on page 7. If the figure is no longer to be included as part of the submission please remove all reference to it within the text.

Reviewers' comments:

Reviewer's Responses to Questions

**Comments to the Author**

1. Is the manuscript technically sound, and do the data support the conclusions?

Reviewer #1: No

Reviewer #2: Yes

2. Has the statistical analysis been performed appropriately and rigorously?

Reviewer #1: I Don't Know

Reviewer #2: Yes

3. Have the authors made all data underlying the findings in their manuscript fully available?

Reviewer #1: No

Reviewer #2: Yes

4. Is the manuscript presented in an intelligible fashion and written in standard English?

Reviewer #1: Yes

Reviewer #2: Yes

5. Review Comments to the Author

Reviewer #1: Leon et al have written a study that used a biostatistical model to analyze infants’ genetic susceptibility to increased risk of dental caries due to C.albicans & Str. mutans collaboration.

The clinical problem is sound and their approach interesting. However, there are severe problems in the article. The involvement and presentation of the biostatistical model is given in a mathematical format. First look, it seemed convincing, with the formulas used. At the closer look, it was very difficult to read, and finally, it was so complicated and specified that I was not able to interpret the method. As a MD, PhD and clinical researcher, I have no such background education or qualification to be able to comment on the method. After considering, I am quite sure that this regards most of the colleague readers as well. Subsequently, perhaps the best forum for this mathematical model is not a clinical medicine publication but a biostatical journal? Therefore, I would recommend that the present paper, or at least the method, should be published in a journal of the correct discipline with mathematically qualified referees. Maybe the clinical part could be rewritten afterwards, by referring that study?

If the article is however meant to be read as a clinical research, there are further concerns:

Abstract:

- The results are not given at all; the authors move straight from the methods to the interpretation.

- The whole description of a clinical study should be written in past tense, i.e., in imperfect, not in the present tense.

Introduction:

- The clinical problem and the previous works were well described.

- Line 49: there is a missing word, or the use of the reference is not correct.

- Tables should be numbered by their appearance in the text; however, here, Table 3 comes first in the text (line 54).

- The authors have set their objective as to explore the potential protective or risk factors; however, I did not find any hypothesis formulated.

- Instead, there is a lengthy explanation (lines 65-81) of how their method should work - I think this kind of promotion should be placed in the discussion part.

Methods:

- Please see above. The mathematical modeling is not adequately placed, requiring specialists on the field, not MDs, to judge it. The replication of the study is not possible.

- Supposing that this research is a clinical, observative study, there are some serious flaws in the methods: the descriptions of the study setting and background population, the selection criteria of the patients, definitions of the primary and secondary outcomes and the conditions studied, ethical permissions, and other statistical methods used, are lacking. No sample size is given, let alone sample size or effect size calculations.

Results:

- It seems that the authors have mixed here some background information and results as well. The paragraph 3.1 should be rewritten, separating them to their adequate places.

- Paragraph 3.2. includes a Fig 1 legend. However, there were no figures within the submission files.

- I understood that the authors have calculated correlation coefficients for some background information. However, the presentation in the tables is not sufficient, e.g., all tables lack patient numbers completely - mere percents are not enough. Confidence intervals are told to be expressed in the tables 2+3, but they are not. Deduction of the most important finding is impossible from the Tables.

Discussion:

- Includes a lot of previous data but not a result summary, deep discussion of the present results, their impact on the clinical work, or the patients. How did this work benefit the studied children?

Reviewer #2: 1. In the Results section, several predictors (e.g., maternal education, plaque index, antifungal use) were selected by the lasso procedure but did not reach statistical significance (Table 2). The manuscript should clearly distinguish between predictors that are statistically significant, those that improve predictive performance, and those with potential clinical implications, to avoid overstating conclusions.

2. The Discussion section alternates between biological, behavioral, and socioeconomic explanations for higher Candida prevalence in Black infants. This should be revised to provide a consistent, evidence-based rationale that integrates vertical and horizontal transmission pathways with social determinants of health.

3. The estimation procedure of the PLSMM model (Section 2.1–2.4) is highly technical and may not be easily reproducible. The authors should supplement the description with intuitive explanations, pseudocode or flowcharts, and additional details about tuning parameter selection, convergence criteria, and software implementation.

4. Table 2 (stratified analysis) shows non-significant race effects, while Table 3 (non-stratified analysis) reports race as a strong predictor (p = 0.0002). The manuscript should explicitly address this discrepancy, clarifying why stratification attenuates the race effect and what this implies about confounding or mediation.

6. PLOS authors have the option to publish the peer review history of their article (what does this mean?). If published, this will include your full peer review and any attached files.

Reviewer #1: No

Reviewer #2: No

---

## [Author Response · Author response to Decision Letter 1]

1 Dec 2025

Response to the Associate Editor

We thank the Associate Editor for handling this paper and providing highly constructive comments. We have carefully considered all the suggested changes, and our responses are detailed below.

1. Please ensure that your manuscript meets PLOS ONE's style requirements, including those for file naming. The PLOS ONE style templates can be found at https://journals.plos.org/plosone/s/file?id=wjVg/PLOSOne_formatting_sample_main_body.pdf and https://journals.plos.org/plosone/s/file?id=ba62/PLOSOne_formatting_sample_title_authors_affiliations.pdf.

We made all the necessary edits to meet PLOS ONE’s style requirements.

2. Thank you for stating the following financial disclosure: “NIH/NIDCR R01DE031025 and NSF SCC 2238208” Please state what role the funders took in the study. If the funders had no role, please state: "The funders had no role in study design, data collection and analysis, decision to publish, or preparation of the manuscript." If this statement is not correct you must amend it as needed. Please include this amended Role of Funder statement in your cover letter; we will change the online submission form on your behalf.

We would like our Funding Statement to be amended to “Leon, Alomeir, Xiao and Wu were partly supported by NIH/NIDCR R01DE031025 and NSF SCC 2238208. The funders had no role in study design, data collection and analysis, decision to publish, or preparation of the manuscript.” The statement was added to the cover letter.

3. In the online submission form you indicate that your data is not available for proprietary reasons and have provided a contact point for accessing this data. Please note that your current contact point is a co-author on this manuscript. According to our Data Policy, the contact point must not be an author on the manuscript and must be an institutional contact, ideally not an individual. Please revise your data statement to a non-author institutional point of contact, such as a data access or ethics committee, and send this to us via return email. Please also include contact information for the third party organization, and please include the full citation of where the data can be found.

Data Availability: Data cannot be shared publicly as participants did not give consent for their data to be shared in this manner. Since the consent statement approved by the Institutional Review Board (IRB) of the University of Rochester, and signed by the participants, did not include the provision that data would be made publicly available, we do not have participant consent to share this data. Requests for anonymized data can be made to the Institutional Review Board (IRB) of the University of Rochester (https://www.rochester.edu/ohsp/irb-review/).

a. If there are ethical or legal restrictions on sharing a de-identified data set, please explain them in detail (e.g., data contain potentially identifying or sensitive patient information, data are owned by a third-party organization, etc.) and who has imposed them (e.g., a Research Ethics Committee or Institutional Review Board, etc.). Please also provide contact information for a data access committee, ethics committee, or other institutional body to which data requests may be sent

b. If there are no restrictions, please upload the minimal anonymized data set necessary to replicate your study findings to a stable, public repository and provide us with the relevant URLs, DOIs, or accession numbers. Please see http://www.bmj.com/content/340/bmj.c181.long for guidelines on how to de-identify and prepare clinical data for publication. For a list of recommended repositories, please see https://journals.plos.org/plosone/s/recommended-repositories. You also have the option of uploading the data as Supporting Information files, but we would recommend depositing data directly to a data repository if possible.

Data Availability: Data cannot be shared publicly as participants did not give consent for their data to be shared in this manner. Since the consent statement approved by the Institutional Review Board (IRB) of the University of Rochester, and signed by the participants, did not include the provision that data would be made publicly available, we do not have participant consent to share this data. Requests for anonymized data can be made to the Institutional Review Board (IRB) of the University of Rochester (https://www.rochester.edu/ohsp/irb-review/).

5. In the online submission form you indicate that your data is not available for proprietary reasons and have provided a contact point for accessing this data. Please note that your current contact point is a co-author on this manuscript. According to our Data Policy, the contact point must not be an author on the manuscript and must be an institutional contact, ideally not an individual. Please revise your data statement to a non-author institutional point of contact, such as a data access or ethics committee, and send this to us via return email. Please also include contact information for the third party organization, and please include the full citation of where the data can be found.

Please see our response to comment 3.

6. Thank you for stating the following in the Acknowledgments Section of your manuscript: “Leon, Alomeir, Xiao and Wu were partly supported by NIH/NIDCR R01DE031025 and NSF SCC 2238208.” We note that you have provided additional information within the Acknowledgements Section that is not currently declared in your Funding Statement. Please note that funding information should not appear in the Acknowledgments section or other areas of your manuscript. We will only publish funding information present in the Funding Statement section of the online submission form. Please remove any funding-related text from the manuscript and let us know how you would like to update your Funding Statement. Currently, your Funding Statement reads as follows: “NIH/NIDCR R01DE031025 and NSF SCC 2238208” Please include your amended statements within your cover letter; we will change the online submission form on your behalf.

We have removed the funding information from the manuscript and added the following statement in the cover letter: “Leon, Alomeir, Xiao and Wu were partly supported by NIH/NIDCR R01DE031025 and NSF SCC 2238208. The funders had no role in study design, data collection and analysis, decision to publish, or preparation of the manuscript.”

7. Please upload a copy of Figure 1, to which you refer in your text on page 7. If the figure is no longer to be included as part of the submission please remove all reference to it within the text.

A copy of Figure 1 has been uploaded.

No specific citations were recommended by the reviewers.

Response to the Reviewers

Reviewer #1:

We thank Reviewer #1 for the very helpful suggestions and comments. We have considered all the points and hope this revision addresses those questions.

Leon et al have written a study that used a biostatistical model to analyze infants’ genetic susceptibility to increased risk of dental caries due to C.albicans & Str. mutans collaboration. The clinical problem is sound and their approach interesting. However, there are severe problems in the article. The involvement and presentation of the biostatistical model is given in a mathematical format. First look, it seemed convincing, with the formulas used. At the closer look, it was very difficult to read, and finally, it was so complicated and specified that I was not able to interpret the method. As a MD, PhD and clinical researcher, I have no such background education or qualification to be able to comment on the method. After considering, I am quite sure that this regards most of the colleague readers as well. Subsequently, perhaps the best forum for this mathematical model is not a clinical medicine publication but a biostatical journal? Therefore, I would recommend that the present paper, or at least the method, should be published in a journal of the correct discipline with mathematically qualified referees. Maybe the clinical part could be rewritten afterwards, by referring that study?

We agree that the Methods section in the previous submission was formula-heavy. To address Reviewer #1’s comment, we have simplified Section 2.2, which previously contained the most formulas, by removing several formulas and instead directing readers to our related statistical journal publication for full procedural and theoretical details. We have also added more intuitive explanations throughout Sections 2.2-2.4 to make the methods more accessible. While we retained some mathematical expressions to preserve essential methodological detail, we believe the revised presentation strikes a balance between completeness and readability for a general audience.

• Abstract:

o The results are not given at all; the authors move straight from the methods to the interpretation.

The results have been added to the abstract.

o The whole description of a clinical study should be written in past tense, i.e., in imperfect, not in the present tense.

We have revised the study description to use past tense.

• Introduction:

o Line 49: there is a missing word, or the use of the reference is not correct.

The missing words “Leon and Wu (2025)” have been added.

o Tables should be numbered by their appearance in the text; however, here, Table 3 comes first in the text (line 54).

We understand that tables are typically numbered according to their order of appearance in the text; however, here we are referring to a table in the appendix. To make this clear, we have changed the reference from Table 3 to Table 1A.

o The authors have set their objective as to explore the potential protective or risk factors; however, I did not find any hypothesis formulated.

It is true that we did not formulate a formal hypothesis in the traditional clinical-trial sense. This is because our paper focuses on introducing a statistical method, along with its new formulation for stratified analyses, demonstrating its application to a clinical study. The data analysis is exploratory by nature as it involves variable selection.

o Instead, there is a lengthy explanation (lines 65-81) of how their method should work - I think this kind of promotion should be placed in the discussion part.

The explanation has been moved to the discussion.

• Methods:

o Please see above. The mathematical modeling is not adequately placed, requiring specialists on the field, not MDs, to judge it. The replication of the study is not possible.

The mathematical descriptions have been simplified, and the method itself has been reviewed by statisticians and published in a statistical journal. This manuscript presents a novel use case of the method, and we believe that the mathematical details provided, along with the software implementation in the R package, are sufficient to allow full reproduction of the study.

o Supposing that this research is a clinical, observative study, there are some serious flaws in the methods: the descriptions of the study setting and background population, the selection criteria of the patients, definitions of the primary and secondary outcomes and the conditions studied, ethical permissions, and other statistical methods used, are lacking. No sample size is given, let alone sample size or effect size calculations.

We note that this study is not a clinical trial or a prospective observational study. Rather, it focuses on a statistical method and its new formulation for stratified analysis. The data analysis is exploratory and it applies the PLSMM with lasso variable selection to existing clinical and demographic data. As such, aspects like patient recruitment, primary or secondary outcomes, and formal sample size calculations are less directly relevant; the emphasis is on identifying potential associations. The credibility of these associations is strengthened by the ability to perform valid statistical inference within this framework, reducing the likelihood that the findings are due to chance.

• Results:

o It seems that the authors have mixed here some background information and results as well. The paragraph 3.1 should be rewritten, separating them to their adequate places.

We have rewritten Section 3.1. The part related to the background was moved to the introduction section.

o Paragraph 3.2. includes a Fig 1 legend. However, there were no figures within the submission files.

The figure has been added.

o I understood that the authors have calculated correlation coefficients for some background information. However, the presentation in the tables is not sufficient, e.g., all tables lack patient numbers completely - mere percents are not enough. Confidence intervals are told to be expressed in the tables 2+3, but they are not. Deduction of the most important finding is impossible from the Tables.

Counts for Black and non-Black groups have been added in Table 1. The confidence intervals have also been added to Tables 2 & 1A.

• Discussion:

o Includes a lot of previous data but not a result summary, deep discussion of the present results, their impact on the clinical work, or the patients. How did this work benefit the studied children?

We have included a summary of the results and the impact of this study in the discussion section.

Reviewer #2:

We thank Reviewer #2 for the very helpful suggestions and comments. We have carefully considered all the points raised and hope that this revision adequately addresses all the concerns.

1. In the Results section, several predictors (e.g., maternal education, plaque index, antifungal use) were selected by the lasso procedure but did not reach statistical significance (Table 2). The manuscript should clearly distinguish between predictors that are statistically significant, those that improve predictive performance, and those with potential clinical implications, to avoid overstating conclusions.

A sentence was added to the third paragraph of Section 3.2 (lines 243-245) to clarify this point and avoid overstating the conclusions. For a detailed discussion, please refer to the last paragraph of Section 3.2 (lines 267-275), which explains that some predictors were selected for their predictive performance but did not achieve statistical significance in the post-selection inference.

2. The Discussion section alternates between biological, behavioral, and socioeconomic explanations for higher Candida prevalence in Black infants. This should be revised to provide a consistent, evidence-based rationale that integrates vertical and horizontal transmission pathways with social determinants of health.

Thank you. We have addressed this point in the discussion section.

3. The estimation procedure of the PLSMM model (Section 2.1–2.4) is highly technical and may not be easily reproducible. The authors should supplement the description with intuitive explanations, pseudocode or flowcharts, and additional details about tuning parameter selection, convergence criteria, and software implementation.

We have substantially revised Sections 2.2-2.4. These sections now contain fewer and simplified formulas, more intuitive explanations, and direct references for further details, including pseudocode. Additionally, we prov

---

## [Decision Letter · Decision Letter 1]

18 Dec 2025

Maternal and Demographic Factors Influencing Oral Candida albicans in Infants: A Stratified Analysis Using a Novel Partial Linear Semiparametric Mixed-effects Model

PONE-D-25-38257R1

Dear Dr. Wu,

We’re pleased to inform you that your manuscript has been judged scientifically suitable for publication and will be formally accepted for publication once it meets all outstanding technical requirements.

Kind regards,

Geelsu Hwang, Ph.D.

Academic Editor

PLOS One

Additional Editor Comments (optional):

Reviewers' comments:

Reviewer's Responses to Questions

**Comments to the Author**

1. If the authors have adequately addressed your comments raised in a previous round of review and you feel that this manuscript is now acceptable for publication, you may indicate that here to bypass the “Comments to the Author” section, enter your conflict of interest statement in the “Confidential to Editor” section, and submit your "Accept" recommendation.

Reviewer #2: All comments have been addressed

2. Is the manuscript technically sound, and do the data support the conclusions?

Reviewer #2: Yes

3. Has the statistical analysis been performed appropriately and rigorously?

Reviewer #2: Yes

4. Have the authors made all data underlying the findings in their manuscript fully available?

Reviewer #2: Yes

5. Is the manuscript presented in an intelligible fashion and written in standard English?

Reviewer #2: Yes

6. Review Comments to the Author

Reviewer #2: The authors have modified the manuscript according to the reviewer's comments completely. All the points have been addressed as well as implemented in the manuscript appropriately. The manuscript is now acceptable.

7. PLOS authors have the option to publish the peer review history of their article (what does this mean?). If published, this will include your full peer review and any attached files.

Reviewer #2: No

---

## [Editor Report · Acceptance letter]

PONE-D-25-38257R1

PLOS One

Dear Dr. Wu,

I'm pleased to inform you that your manuscript has been deemed suitable for publication in PLOS One. Congratulations! Your manuscript is now being handed over to our production team.

Kind regards,

on behalf of

Dr. Geelsu Hwang

Academic Editor

PLOS One